# Utility of the Specialized Pro-Resolving Mediators as Diagnostic and Prognostic Biomarkers in Disease

**DOI:** 10.3390/biom12030353

**Published:** 2022-02-23

**Authors:** Jesmond Dalli, Esteban Alberto Gomez, Charlotte Camille Jouvene

**Affiliations:** 1William Harvey Research Institute, Barts and The London School of Medicine and Dentistry, Queen Mary University of London, Charterhouse Square, London EC1M 6BQ, UK; e.a.g.c.cifuentes@qmul.ac.uk (E.A.G.); charlotte.jouvene@gmail.com (C.C.J.); 2Centre for Inflammation and Therapeutic Innovation, Queen Mary University of London, London EC1M 6BQ, UK

**Keywords:** omega-3 fatty acids, resolvin, protectin, maresin, lipoxin, chronic inflammation

## Abstract

A precision medicine approach is widely acknowledged to yield more effective therapeutic strategies in the treatment of patients with chronic inflammatory conditions than the prescriptive paradigm currently utilized in the management and treatment of these patients. This is because such an approach will take into consideration relevant factors including the likelihood that a patient will respond to given therapeutics based on their disease phenotype. Unfortunately, the application of this precision medicine paradigm in the daily treatment of patients has been greatly hampered by the lack of robust biomarkers, in particular biomarkers for determining early treatment responsiveness. Lipid mediators are central in the regulation of host immune responses during both the initiation and resolution of inflammation. Amongst lipid mediators, the specialized pro-resolving mediators (SPM) govern immune cells to promote the resolution of inflammation. These autacoids are produced via the stereoselective conversion of essential fatty acids to yield molecules that are dynamically regulated during inflammation and exert potent immunoregulatory activities. Furthermore, there is an increasing appreciation for the role that these mediators play in conveying the biological actions of several anti-inflammatory therapeutics, including statins and aspirin. Identification and quantitation of these mediators has traditionally been achieved using hyphenated mass spectrometric techniques, primarily liquid-chromatography tandem mass spectrometry. Recent advances in the field of chromatography and mass spectrometry have increased both the robustness and the sensitivity of this approach and its potential deployment for routine clinical diagnostics. In the present review, we explore the evidence supporting a role for specific SPM as potential biomarkers for patient stratification in distinct disease settings together with methodologies employed in the identification and quantitation of these autacoids.

## 1. Introduction

Recent years have seen an increased appreciation for the need of developing a more personalized approach to the treatment of patients with inflammatory conditions [1,2]. Indeed, it is now abundantly apparent that the paradigm where patients are treated using a broad and formulaic approach does not best serve them. This is because such an approach leads to delays in their treatment with effective therapeutics, thus allowing the condition to precipitate. In addition, the administration of ineffective drugs also unnecessarily exposes these patients to their unwanted side effects [1,2]. Such considerations are especially important in conditions where there is a progressive destructive inflammation, such as, for example, in patients with rheumatoid arthritis (RA) treated with disease modifying anti-rheumatic drugs, where up to 50% of patients will not respond to these therapeutics [3,4]. Several aspects have hampered the development of patient tailored therapeutics, including a limited understanding of the molecular and cellular mechanisms that contribute to the disease as well as the availability of robust biomarkers that are effective at predicting treatment responses or sufficiently sensitive for evaluating treatment responsiveness early in the treatment regime.

Prostaglandins, thromboxanes and leukotrienes are initiators of the inflammatory response promoting vascular leak, leukocyte recruitment, thrombosis and smooth muscle contraction. Lipoxins, resolvins, protectins and maresins are agonists of resolution. They counter-regulate the formation and actions of inflammation-initiators and promote the clearance of apoptotic cells, the uptake and killing of bacteria by phagocytes and tissue repair and regeneration.

Studies conducted in the early 2000s investigating mechanisms engaged by the immune system to regulate innate immune cell responses and trafficking during acute inflammation uncovered a novel genus of mediators produced from essential fatty acids (Figure 1). These autacoids were characterized as specialized pro-resolving mediators (SPM) given their potent biological activities in curtailing inflammation [5,6,7]. SPM are produced via the stereoselective conversion of essential fatty acids by specific biosynthetic enzymes [8,9]. Production of these autacoids is initiated by lipoxygenases (encoded by the ALOX genes), primarily ALOX5, ALOX12, ALOX15 and cyclooxygenase (COX)-2 [6,7,8,9,10,11,12,13,14,15,16]. Among the first described SPM were the eicosapentaenoic acid (EPA) and docosahexaenoic acid (DHA)-derived resolvins, termed as E-series (RvE) and D-series resolvins (RvD), respectively [6,7]. More recently we found that, in addition to arachidonic acid (AA), EPA and DHA, n-3 docosapentaenoic acid (n-3 DPA) is also substrate in the formation of novel SPM that share biosynthetic pathways with the DHA-derived resolvins, protectins and maresins [11,12]. Table 1 provides a summary of the mediators that form part of the distinct SPM families together with their complete stereochemistries where established.

An extensive amount of work has detailed the biology of these molecules in governing inflammation, demonstrating that each of these autacoids is able to regulate the production of pro-inflammatory mediators (including cytokines and eicosanoids), leukocyte trafficking to the site of inflammation, the ability of phagocytes to uptake bacteria and the ability of macrophages to clear apoptotic cells [8,17,18,19,20,21,22,23,24,25]. Mounting evidence demonstrates SPM are central in regulating adaptive immunity, including T-cell effector function [26,27,28], regulatory T-cell differentiation [24,28] and B-cell antibody class switch [29,30]. Furthermore, the biology of these molecules extends beyond the regulation of immune cell responses, with several studies detailing the ability of these autacoids in promoting epithelial cell migration [31,32], the production of bactericidal proteins by epithelial cells [33,34] and the regulation of endothelial cell responses [35,36] among others. Notably, their biological activities are mediated by cognate receptors that form part of the G-protein-coupled receptor family [8,37]. Deletion of these receptors or loss of SPM production have been linked with the exacerbation of inflammation in several experimental disease settings [38,39]. Together, this wealth of knowledge underscores a central role of SPM in governing both immune and stromal cell responses to limit inflammation.

Further supporting a role for SPM in the onset and/or progression of inflammatory disorders are studies conducted by several laboratories in a wide range of human disease settings, including rheumatoid arthritis [40,41], periodontal disease [42] and atherosclerosis [43,44], demonstrating that SPM production and/or activity is dysregulated. Intriguingly, several studies have also detailed a role for these molecules in mediating the biological activities of a number of commonly used drugs, such as aspirin [45,46] and statins [11,47,48].

As with many small molecules, the state-of-the-art methodologies employed in the identification and quantitation of SPM are those based on liquid chromatography tandem mass spectrometry. This approach provides the distinct advantage of being able to simultaneously measure the concentrations of multiple molecules originating from different types of biological tissues with a relatively high degree of sensitivity and specificity. Liquid chromatography is an essential step in the analytical process, given that many of the SPM are either positional or chiral isomers of each other. Using reverse phase chromatography one can separate many of these species (Figure 2). Intriguingly, reverse phase chromatography using C18 columns is not only able to separate positional isomers but also chiral isomers of the SPM. This is in line with what is observed for other lipid mediators, such as leukotriene B_4_, whereby, using reverse phase chromatography, one can easily separate both the double bond and positional isomers of this molecule [49]. Mediator separation using reverse phase chromatography can be achieved using both high pressure liquid chromatography (HPLC) and ultra-high pressure liquid chromatography (UPLC) systems.

Identification and quantitation of SPM is achieved using tandem mass spectrometers. There is a range of instrumentation employed for the quantitation of lipid mediators that take advantage of different technologies, including different ionization and detector technologies. The analysis of SPM is generally dependent on electro spray ionization and the use of multiple reaction monitoring (MRM) methodologies. This relies on the use of pre-developed methodologies with pre-set transitions for each of the mediators of interest, where the precursor ion and a characteristic product ion are monitored for each molecule. In the elaboration of such methodologies, great care must be taken to ensure that the instrument parameters being used are carefully tuned for each molecule and for each of the transitions, given that in many instances even minor changes in parameters, such as collision energies, may result in marked changes in sensitivity. The advantage of using a targeted approach with pre-defined MRM transitions is that instrument sensitivity can be increased by using a schedule for when each of the transitions should be monitored, thereby reducing the time that the instrument spends on the acquisition of data in regions of the chromatogram which are not of interest. Given the strong reliance on MRM methodologies, and the fact that many of the molecules being monitored are positional isomers, triple quadrupole mass spectrometers are extensively used in the study of SPM. Given the expanding utility and improvements in available instrumentation and methodologies, in the present review we will discuss the evidence supporting the utility of distinct SPM as diagnostic and/or predictive biomarkers in a range of inflammatory conditions.

## 2. Additional Quantitation Methodologies

While targeted LC-MS/MS remains the gold standard approach for the identification and quantitation of SPM, other approaches have been used for the quantitation of lipid mediators. These include ultraviolet (UV)-HPLC and ELISAs. UV-HPLC based methodologies have a key limitation which is related to the sensitivity of the UV, thereby limiting its use to those tissues and biological systems where SPM concentrations are high. Furthermore, drug metabolites that display similar chromatographic behavior may also interfere with the detection of these molecules, thereby further limiting the application of this approach. ELISAs have been used for the measurement of lipid mediators, including leukotrienes and prostaglandins, in biological systems for many decades. In recent years, ELISAs have been developed for the measurement of SPM which are widely employed in the measurement of these molecules in human tissues and experimental systems. These types of assays allow many samples to be analyzed simultaneously at relatively low cost, thereby expanding its potential utility in diagnostics. One aspect that has been viewed unfavorably by some investigators is the specificity of this methodology to identify the target mediator without interference from matrix or other isomers. Another aspect to be mindful of is that ELISAs should be used with caution, and investigators should ensure that ELISAs being used for the quantitation of SPM have been appropriately validated to minimize the likelihood of erroneous identification/quantitation of these autacoids in biological systems. It should be noted that studies comparing results obtained from ELISAs with those obtained using mass spectrometry suggest that whilst the reported absolute amounts usually differ, the relative quantities and, most importantly, the direction of change (i.e., upregulation or downregulation) observed between different study groups remain similar. This aspect is encouraging since it suggests that with the appropriate benchmarking and method development, ELISAs might prove to be useful in the development of scalable diagnostics. This aspect is further enhanced by recent technological developments including the development of aptamer technologies [50] and small molecule arrays [51].

## 3. Sample Preparation and Quantitation Methodologies

Given the relatively low abundance of these molecules, it is essential that samples are extracted from their biological matrix using solid phase extraction methodologies. The main methodologies employed for this process involve reverse phase C18 columns. Normal phase (silica) and ion exchange (anion or cation) phase have also been applied for the extraction of SPM and other lipid mediators with different phases, providing select advantages in both limiting matrix carryover and the extraction of specific lipid mediator species based on their hydrophobicity and ion-exchange capacity [52]. Identifying suitable extraction methodologies that minimize matrix carryover and ensure maximal recovery of these molecules is a critical step given that matrix interference will markedly influence the sensitivity of the assay and potentially result in false negative results.

In order to facilitate the identification and quantitation of SPM, isotope labelled internal standards are added to the samples prior to extraction. The most widely used isotope labelled standards in the study of lipid mediators are those labelled using deuterium labels. The use of this isotope results in molecules that display essentially similar chromatographic and fragmentation profiles but with a distinct molecular mass, meaning that they can be easily distinguished from the biological forms. These molecules are used to monitor inter-run changes in the expected retention times of the biological molecules that may result from matrix-associated changes in the ability of the molecules to interact with the stationary phase or technical issues. Furthermore, because these standards are added prior to sample extraction, their absolute quantities in each of the samples can be used to estimate sample recoveries, thereby facilitating the quantitation of the biological molecules in each of the samples. To achieve this, isotope labels representing distinct regions of the chromatogram and distinct carbon lengths and number functional groups (e.g., hydroxyl groups or peptide moieties) are employed as surrogates to calculate recoveries [40,53]. The limitation of this approach is that, to date, only a limited number of internal standards are available and therefore one cannot precisely calculate the absolute concentrations of those molecules where the corresponding isotope labelled standard is not available. Therefore, caution must be taken in the interpretation of such quantitative data, especially for those molecules where the corresponding isotope labelled molecules is not available since the absolute amounts may not necessarily correspond to the calculated amount using this surrogate approach.

## 4. Diagnostic Utility of SPM in Infections

Studies investigating the biology of SPM in the context of bacterial infections have elucidated a role for these mediators in regulating host immunity. For example, in several experimental systems, resolvin (Rv) D1 and its 17R-epimer exert protective activities against several bacterial pathogens, including *Citrobacter rodentium* [54], *Escherichia coli* [55,56] and *Staphylococcus aureus* [55], via the regulating host immune response to downregulate the amplitude of the inflammatory process while enhancing the ability of phagocytes to clear the invading pathogens. These biological actions were also observed in models of polymicrobial sepsis where SPM, including RvD2 and Maresin (MaR) 1 [57], limit phagocyte recruitment to the site of infection, reduce the production of local and systemic inflammatory mediators, promote bacterial clearance and enhance survival. These biological activities of SPM were also recently found to be retained in humans. Using an elegant model of dermal bacterial infection, Motwani et al. demonstrated that local administration of SPM, namely LXB_4_, RvE1, RvD2 and 17R-RvD1, at concentrations equivalent to those measured in the tissues during active infection accelerated the resolution of inflammation [58].

Given these remarkable host protective activities of SPM in facilitating the clearance of bacteria, the question of whether the endogenous levels of these molecules might shed light on disease status in humans has arisen. In this context Norris et al. found that SPM production was downregulated in healthy volunteers following LPS challenge [59]. Intriguingly, this dysregulation of SPM biosynthesis was observed even in volunteers that were previously administered omega-3 supplements and recovers with time after challenge. These observations suggest that exposure of the host to microbial components might disrupt SPM production, facilitating disease. Studies with septic patients support the observation that SPM production is dysregulated during infections [60]. Indeed, here it was observed that plasma SPM concentrations, together with concentrations of classic eicosanoids, were linked with 28-day mortality. Interestingly, this distinction between patients that survived and those that did not was observed to arise as early as 24 h after admittance to the Intensive Care Unit. This observation that SPM concentrations might be linked with patient outcome was also made in patients with meningeal tuberculosis [61]. As a result, overall levels of these mediators in the cerebrospinal fluid of patients with meningeal tuberculosis were markedly reduced in those patients that subsequently succumbed to the disease when compared with those that survived.

The biological actions of SPM are also reported to extend into the regulation of host immunity during viral infections. For example, the DHA-derived RvD1 and its precursor 17-hydroxy-docosahexaenoic acid (17-HDHA) regulate B-cell response during H1N1 infections in mice by promoting antibody class switch [62]. The Protectin (PD) family of mediators regulate viral propagation by inhibiting intracellular viral RNA transport mechanisms [63]. Similarly, the EPA-derived RvE1 reduces effector T-cell and neutrophil-mediated propagation of inflammation during herpes simplex virus infections [64]. Recent studies also suggest a role of these mediators in regulating host immune responses to severe acute respiratory syndrome corona virus (SARS-CoV)-2. In these, RvD1 and RvD2 govern the production of inflammatory cytokines by macrophages, including IL-8 and TNF-a, via the regulation of micro-RNA expression [65]. Studies in humans demonstrate that SPM levels in either broncho alveolar lavage fluids or serum are linked with disease outcome in viral infections. Archambault et al. found that D-series resolvins and LXA_4_ concentrations, together with levels of several pro-inflammatory eicosanoids were increased in bronchoalveolar lavages of intubated COVID-19 patients [66]. Similar observations were also made in serum from patients with COVID-19, where levels of these autacoids were elevated with increased disease severity [67]. Thus, these findings support a potential role for measuring SPM concentrations in patients with both bacterial and viral infections to aid in the management of these patients by establishing early in the course of their hospitalization their disease trajectory and therefore design optimal treatment strategies.

## 5. Pro-Resolving Mediators as Biomarkers in Chronic Inflammatory Conditions

Amongst the key biological activities of SPM is their ability to limit leukocyte infiltration to the site and counter-regulate the formation of pro-inflammatory mediators [68]. Thus, there is extensive interest in establishing whether the production of these molecules is disrupted in chronic inflammatory conditions and whether the pharmacological administration of SPM may represent a useful approach for treating such conditions. Indeed, studies conducted by many groups demonstrate that SPM production becomes dysregulated in chronic inflammatory settings and that the use of these mediators or their analogues may limit disease progression and in some instanced even reverse some of the damage. Table 2 provides a non-exhaustive summary of studies reporting on SPM levels in human disease and their regulation by supplementation strategies. Interested readers are also referred to a recent comprehensive review detailing the biological actions of SPM in a range of disease setting [8]. In the remainder of this section, we will discuss some of the evidence related to a select group of these conditions. We also direct interested readers to other, more comprehensive reviews that detail aspects of SPM biology that are not covered below [8,21,69,70,71,72,73]:

### 5.1. Periodontal Disease

Periodontal disease is characterized by unremitting inflammation that results in significant damage to the periodontium. Studies in experiential settings demonstrate that SPM exert potent activities in the regulation of periodontal inflammation by limiting leukocyte trafficking to the site and downregulating the production of pro-inflammatory mediators. These molecules also promote bone and tissue repair in several models of periodontal disease [36,42,44]. In recent studies found that levels of LXA_4_, PD1 and MaR1 in salivary tissues of patients with periodontal inflammation were correlated with disease severity. Intriguingly, while LXA_4_ levels were observed to be negatively correlated with disease severity, those of PD1 and MaR1 were positively correlated. These results suggest that SPM biosynthetic pathways, or potentially their degradation pathways, are differentially regulated in disease settings likely as an attempt of the host immune response to counter the ongoing inflammatory processes [82].

### 5.2. Rheumatoid Arthritis

This condition, as with periodontal disease, is characterized by unremitting inflammation that leads to progressive tissue destruction. Both the innate and adaptive arms of the immune response are implicated in the pathogenesis of RA. Studies in pre-clinical models demonstrate that several SPM, including RvD1 [95], RvD3 [96] and MaR1 [97], regulate disease progression. In addition, MaR1 was recently found to limit nociceptive hypersensitivity linked with inflammatory arthritis [98]. In humans, Barden et al. observed that synovial RvE2 concentrations were negatively associated with pain score whereas overall plasma SPM concentrations were negatively correlated with erythrocyte sedimentation rate, a marker of disease activity [41]. Separate studies report that serum concentrations of LXA_4_, RvD1 and RvE1 are significantly reduced in patients with active RA when compared with those in remission [99]. Furthermore, plasma SPM concentrations were observed to reflect synovial disease phatotype in RA patients [40]. This observation is of interest since synovial disease phototypes were recently observed to be linked with a differential response to disease-modifying anti-rheumatic drugs [40]. Furthermore, current approaches to stratifying patients using their pathotype are somewhat cumbersome since they involve ultrasound guided biopsies. Therefore, blood SPM concentrations might provide a useful alternative for patient stratification.

### 5.3. Vascular Disease

Atherosclerosis is characterized by the systemic activation of immune cells that are then recruited into the vascular wall and contribute to the accumulation of lipids. Studies in experimental systems demonstrate that loss of ALOX15, a key enzyme in SPM production, was linked with increased vascular lipid load in atherosclerosis prone mice [100]. Pharmacological studies demonstrate that SPM regulate various aspects of disease development. Whereby, 15-epi-LXA_4_ and RvE1 were observed to limit the migration of human saphenous vein SMCs and decrease phosphorylation of the platelet-derived growth factor receptor-β [101]. RvD1 was also observed to exert protective activities in regulating atheroprogression by improving the uptake of apoptotic cells within atherosclerotic lesions, reducing lesional oxidative stress and increasing the thickness of the fibrous cap. RvD2 and MaR1 regulate disease progression in experimental settings this time by reducing macrophage accumulation in vascular tissues and increasing the number of smooth muscle cells [102]. By contrast, RvD5_n-3 DPA_ decreased disease progression by limiting platelet leukocyte interactions [103]. 

SPM levels are also observed to be altered in humans where the concentrations of RvD1 are markedly reduced in vulnerable regions from human carotid atherosclerotic plaques [81]. Plasma levels of 15-epi-LXA_4_ are significantly lower in patients with symptomatic peripheral artery disease than in healthy volunteers [101]. Similarly, peripheral blood n-3 DPA-derived resolvin (RvD_n-3 DPA_) concentrations are lower in patients with cardiovascular disease when compared with those observed in healthy volunteers [103]. This reduction in RvD_n-3 DPA_ concentrations is linked with an increased activation of circulating phagocytes [103], suggesting that these molecules might be useful indicators of vascular disease.

### 5.4. Allergic Inflammation

The biological activities of SPM are also described in allergic conditions including allergic airway diseases such as asthma and allergic rhinitis. In recent studies, RvE3 was observed to reduce the total numbers of inflammatory cells and eosinophils recruited into the lung of mice sensitized and challenged with house dust mite. This mediator also reduced the levels of IL-23 and IL-17 in lavage fluid and suppressed the expression of IL-23 and IL-17A mRNA expression in lung and peribronchial lymph node. Importantly, these cellular and molecular changes were linked with a reduction in lung resistance in mice treated with RvE3 [104]. In a murine model of allergic asthma, RvE1 promoted a decrease in leukocyte recruitment into the lung and downregulated the expression of pro-inflammatory cytokines in lavage fluids and in macrophages [105]. In separate studies, the DHA-derived RvD1 and AT-RvD1 were observed to exert protective activities in murine models of asthma. Here, RvD1 markedly decreased airway eosinophilia and mucus metaplasia and downregulated the expression of IL-5 and IκBα degradation. AT-RvD1 also regulated lung inflammation in these settings, leading to a marked decrease in the resolution interval for lung eosinophilia and the expression of inflammatory peptide and lipid mediators, actions that together contributed to an accelerated resolution of airway hyperreactivity to methacholine [106]. 

The protective activities of SPM are also observed with phagocytes from patients with asthma whereby AT-RvD1 reduced TNF-α levels in peripheral blood mononuclear cells from patients with severe asthma stimulated with lipopolysaccharide or *Dermatophagoides pteronyssinus*. This SPM also increased phagocytosis of apoptotic neutrophils by monocytes from patients with severe asthma [107]. PD1 was recently observed to display protective activities in regulating eosinophil biology, a key cell type linked with the propagation of allergic inflammation. Indeed, this SPM limited chemotaxis of these cells towards CCL11/eotaxin-1 and 5-oxo-eicosatetraenoic acid. PD1 also modulated the expression of the adhesion molecules CD11b and L-selectin on these cells [108].

### 5.5. COVID-19

The potent bioactions of SPM in regulating both innate and adaptive immune responses, together with a mounting body of evidence suggesting that the production of these mediators may be dysregulated in hospitalized patients with COVID-19, have sparked interest in evaluating whether these molecules could provide novel leads in treating the excessive inflammatory response observed in these patients. While the direct evidence on the therapeutic potential of these molecules in COVID-19 is at present limited, studied performed thus far in these patients, as well as in other respiratory viral infections such as H1N1, are encouraging. Indeed, in recent studies we found that the incubation of phagocytes from COVID-19 with SPM rectifies their abilities to uptake and kill bacteria. These mediators also downregulate the expression of activation markers on circulating phagocytes, including tissue factor (CD142), which was recently linked with an increased risk of thrombosis in these patients [109]. Furthermore, RvD2, RvD3, MCTR3 and PCTR3 downregulated the expression of inflammatory cytokines in monocyte-derived macrophages from patients with COVID-19, including IFNγ and TNF-α [53]. In murine studies with influenza infections, the protectin family of mediators was observed to both regulate the inflammatory response and limit viral replication by inhibiting the viral export machinery [63]. By contrast, the D-series resolving precursor 17-HDHA increased antigen-specific Ab titers to pH1N1 virus. 17-HDHA also increased the number of antigen-specific antigen-secreting cells present in the bone marrow. Intriguingly, the 17-HDHA-mediated-increased antigen production was more protective against live pH1N1 influenza infection in mice [62]. Together, these findings suggest that SPM may hold promise as new therapeutics for COVID-19 by reprogramming both the innate and adaptive arms of the immune response to decrease inflammation and enhance anti-viral immunity.

## 6. SPM as Biomarkers for Determining Therapeutic Efficacy of Anti-Inflammatory Drugs

Studies investigating the mechanisms of action of several widely used therapeutics such as statins and aspirin suggest a role for SPM in mediating their anti-inflammatory activities (Figure 3). For example, aspirin, in addition to inhibiting prostaglandin and thromboxane formation, promotes the formation epimeric forms of the resolvins, protectins and lipoxins via the acetylation of COX-2 [7,45]. Indeed, this reaction ablates the ability of the enzyme to catalyze the formation of PGG_2_ while retaining its ability to oxygenate AA on carbon 15 and DHA on carbon 17. These mediators display similar binding affinities to the cognate receptor for the ALOX-derived molecules while displaying enhanced stability to metabolic inactivation [110]. The ability of aspirin, in particular low dose aspirin, to upregulate the formation of these molecules was also established in humans, whereby Chiang and colleagues demonstrated in a randomized trial that low dose aspirin, at variance to high dose aspirin, upregulated plasma 15-epi-LXA_4_ (also referred to as AT-LXA_4_) concentrations [111]. Intriguingly, the ability of aspirin to increase 15-epi-LXA_4_ concentrations in healthy volunteers was observed to be dependent on gender and age. Where, a positive correlation was observed in females with age, whilst a negative correlation was found in males in the ability of aspirin to increase plasma 15-epi-LXA_4_ [112]. The gender-specific differences in AT-SPM regulation appear to be tissue or potentially condition-specific. In a separate study, the regulation of 15-epi-LXA_4_ by aspirin in gingival crevicular fluid of patients with periodontal disease was not observed to be different between males and females [113]. These findings suggest that while aspirin may regulate the formation of AT-SPM in distinct tissues; factors such as age, gender and disease may have an influence on the individual mediators being regulated (i.e., 15-epi-LXA_4_ vs. 17R-RvD1 vs. 17R-PD1) and the direction of change (i.e., upregulated or downregulated). Thus, future studies will need to establish which of these autacoids is diagnostic of aspirin efficacy in specific patient populations and target tissue or fluid (e.g., plasma).

Another class of drugs that regulates SPM formation is the statins. In experimental lung inflammation, Lovastatin increases 15-epi-LXA_4_ concentrations via the upregulation of 14,15-epoxyeicosatrienoic acid by airway epithelial cells [47]. Atorvastatin was observed to also upregulate 15-epi-LXA_4_ levels in the heart, albeit via a different mechanism to that observed in the lung. Indeed, in this context atorvastatin regulates cyclooxygenase-2 and 5-lipoxygenase activity to increase 15-epi-LXA_4_ levels [48]. We recently observed that, in vascular endothelial cells, atorvastatin also regulates COX-2 activity by promoting nitrosylation of the enzyme, a reaction that was dependent on the activity of the nitric oxide synthase [11]. This mechanism contributed to the upregulation of 13-series resolvins (RvT) that display potent immune regulatory activity on phagocytes. Indeed, pharmacological inhibition of COX-2 activity blocked RvT formation and reversed the protective activities of atorvastatin in both experimental joint inflammation and bacterial infections [11,114].

Recent studies suggest that dexamethasone may also upregulate SPM formation [115]. In experimental allergic inflammation levels of the DHA-derived protectins and those of resolvin pathway marker 17-HDHA were increased by this corticosteroid. We also observed that this regulation of SPM levels by this potent anti-inflammatory drug was retained in human airway inflammation, with levels of several SPM, including the DHA-derived PD1 and the n-3 DPA-derived MaR1_n-3 DPA_, being markedly upregulated in plasma of patients with COVID-19 treated with dexamethasone [53]. Intriguingly, these observations were linked with the upregulation of several SPM biosynthetic enzymes in circulating leukocytes from patients treated with dexamethasone [53]. These observations suggest that dexamethasone increases SPM formation by regulating the expression of their biosynthetic enzymes.

## 7. Potential for the Use of SPM as Biomarkers for Determining the Utility of Omega-3 Supplements in Regulating Inflammation

Omega-3 supplements have long been held to exert beneficial actions in the regulation of inflammation with a plethora of studies in experimental systems supporting their utility in a range of inflammatory conditions, including cardiovascular disease and arthritis [70,116,117]. The evidence in humans has been less clear-cut, with studies observing both beneficial and potentially harmful effects or no effects. One should note that studies in humans have used a wide range of doses as well as forms of these fatty acids and the influence of these variables on their bioavailability, amongst others, are not fully understood. Given that omega-3 fatty acids are substrates in the formation of SPM, several groups have investigated whether the beneficial actions of these supplements are at least in part linked with the upregulation of these protective autacoids. Studies conducted in humans appear to support this hypothesis. In patients with minor cognitive impairment, the administration of an omega-3 supplement increased plasma RvD1 concentrations and improved cognitive function in these patients [118]. Of note, the epimeric form of RvD1, 17R-RvD1, which exerts its biological actions via the same receptors, was previously reported to also limit post-operative cognitive decline in an experimental model of surgical-induced cognitive decline [119]. In patients with chronic kidney disease, supplementation with omega-3 fatty acids led to an increased production of SPM, primarily RvE1, RvE2, RvE3 and RvD5, by isolated peripheral blood neutrophils. This increase in SPM production was linked with a decrease in myeloperoxidase levels, a marker of neutrophil activation, in plasma from patients receiving omega-3 fatty acids [120].

Recent studies have also started to address the pharmacokinetics of omega-3 supplements by using SPM as functional biomarkers. In studies performed in healthy volunteers, plasma SPM concentrations reached a maximum as early as 2 h after the oral ingestion of the supplement [121]. This increase was also dependent on the initial dose of omega-3 supplement provided, establishing a link between substrate availability and conversion to SPM. Furthermore, in these studies plasma levels of several SPM were correlated with changes in both peripheral blood phagocyte and platelet activation. Of note, plasma levels of these autacoids also rapidly decreased in peripheral blood of healthy volunteers given a single dose of omega-3 supplement, returning to baseline values between 2–24 h after supplementation depending on the supplement dose administered. This observation is in line with the biology of SPM, whereby, as autacoids, these molecules are further metabolized and cleared from the tissue. Experimental evidence obtained so far on the kinetics of these processes has primarily focused on in vitro systems, demonstrating that the rates of conversion of these autacoids varies between different molecules [110,122,123]. Furthermore, some of the further metabolites described for these molecules retain the biological activities of the parent molecule [110,122,123]. These aspects will need to be taken into consideration when devising strategies to monitor supplement efficacy in patient populations. 

Notably, in instances where supplements are administered over longer periods to those employed in our recent studies, there may be some flexibility regarding the timing employed between supplementation and sample collection. In this context, we recently evaluated plasma SPM concentrations in patients with peripheral artery disease. These patients were administered the supplement for 7 days and then blood was collected. Here, plasma SPM levels of these autacoids were elevated several hours after the administration of the last supplement dose compared to pre-supplement levels. These findings suggest that chronic administration may result in sustained substrate release, likely from esterified pools that can support SPM formation for longer periods to those achieved with a single equivalent supplement dose. Furthermore, plasma SPM concentrations in patients with periphery artery disease were increased in dose-dependent manner, an increase that was linked with changes in peripheral blood phagocytes response [124]. Together these observations lend support to the potential utility of peripheral blood SPM as functional biomarkers in determining the efficacy of supplementation at regulating inflammation.

## 8. Utility of Machine Learning in Biomarker Identification

Available evidence from both experimental systems and animal models suggest that distinct mediators are regulated in a context, tissue and condition-specific manner. Therefore, methodologies for identifying and corroborating the potential utility of individual SPM as potential biomarkers in each disease setting are essential in facilitating their clinical application.

Identifying biomarkers by a conventional approach, meaning identifying a candidate and running tests to validate it, requires a great amount of biological evidence and is time-consuming, expensive and risky. To date, new techniques to generate and collect large amounts of data, such as mass spectrometry, have been used in biological and medical research, including drug discovery and biomarker development, for the analysis of a high number of molecules improving the chances of identifying biomarkers of interest [125].

The value of data generated by mass spectrometry depends on the analysis strategies employed. Machine learning methodologies, algorithms to build mathematical models based on patterns and interferences that then can make decisions [126] have been applied to metabolomics data to diagnose illness and, in a more advanced application, to classify and predict outcomes of different diseases and therefore contribute to the identification of biomarkers associated with them. These biomarkers can be used to determine the efficacy of pre-existing treatments and diagnostic strategies [126,127,128].

Biological and medical data are complex, containing multiple layers of information that are usually incomplete, heterogeneous, noisy and present unwanted variability (e.g., operator and batch effects) [128]. Different from univariate association analyses, machine learning can address some of the issues associated with biological data while also providing strategies for considering unknown associations between the different variables present in the samples [129].

Machine learning strategies work as a flow of knowledge: the model is built by learning from sample data, known as training data, and then applies the knowledge to make predictions or classify data [126]. Three steps must be followed for good practice in machine learning application: training, in which the model uses the training dataset to create the classification model, validation, in which the performance of the model is assessed by reshuffling of the training data, and evaluation, which consists of evaluating the model using a new dataset (Figure 4).

Different supervised machine learning methodologies have been used to identify biomarkers through the years, all of them with their pros and cons. Among the most popular are Bayesian classifiers [130], artificial neural networks [131], support vector machine [132] and random forest [133]. The most common disadvantages of these strategies are that they require a great number of samples per group, overfitting (the model is only good to classify the training data), mishandling of multicollinearity (models not identifying correlations between different variables) and sensitivity to the presence of outliers.

When applied correctly, machine learning helps to understand complex biological data without missing unknown interactions between the features and leading to the identification of relevant biomarkers in different conditions. For this to occur, the training data needs to be strong enough (a large number of samples and features) to create reliable models. A pre-processing step derived from exploratory analysis can be helpful to make the data stronger by identifying underlying correlations, taking out outlier values and scaling the features. Choosing the best machine learning methodology is always fundamental and it will depend on the available data and the biological question that needs to be answered. It is important to consider the disadvantages of each methodology so that measures can be taken. Finally, evaluating the models in new and independent datasets is the most important step if the purpose of the machine learning strategy is to create a model able to predict between classes, identify biomarkers and give an insight into the biological processes behind the prediction logic.

We recently applied these models to evaluate whether they can be utilized to identify biomarkers that are prognostic of treatment responsiveness. Using machine learning, we devised models that were trained using plasma lipid mediator concentrations of patients that responded to disease modifying anti-rheumatic drugs, front line therapeutics in rheumatoid arthritis and plasma from patients that did not respond [40]. These models identified candidate mediators that were then validated using plasma from a second cohort of patients, demonstrating that these biomarkers were prognostic of treatment responsiveness with an accuracy in excess of 85% [40]. Due to the relatively small number size, this initial study used data obtained from both males and females for both model building and validation. Future studies will need to evaluate whether the sensitivity and accuracy of these models can be further enhanced by evaluating lipid mediator concentrations from different sexes separately in light of findings suggesting that sex and sex hormones influence lipid mediator production [134,135]. Nonetheless, given that patient responsiveness to a whole host of therapeutics is acknowledged to vary and may expose patients to unwanted side effects of given therapeutics, prognostic biomarkers may be a game changer in the development of a personalized medicines approach that is better suited to the individual patient.

## 9. Current Limitations and Future Directions

Evidence presented in previous sections supports the potential utility of specific SPM as biomarkers in distinct disease settings. It should be emphasized that given the dynamic regulation of these autacoids and the evidence available thus far, it is likely that there will not be one or a small number of SPM that may be universally useful as biomarkers in a wide range of disease settings. Indeed, it is likely that one or a panel of SPM may be only useful in select disease settings, and likely only for a given tissue or biological sample. This aspect provides both advantages, in that one can have specific diagnostic tests for given conditions, as well as disadvantages, in that tests will need to be developed and validated for every application, thereby increasing the costs. One also has to recognize that available evidence will still need to be validated in prognostic studies, ideally in a multi-center setup. This will help to establish both the robustness as well as the generalizability of potential biomarkers, especially in light of recent studies that suggest a differential regulation of lipid mediators in different ethnic groups [78]. Another aspect that will require careful consideration is platforms to be used for the analysis of these molecules. It is widely appreciated that there are marked differences in the ability of different mass spectrometry platforms to identify small molecules, and in particular low abundance molecules such as lipid mediators. Therefore, bench marking of instrumentation and methodologies using standard reference materials will be essential in order to ensure that these are sufficiently sensitive to quantify these molecules in a robust manner. Finally, current methodologies for sample preparation and mediator identification are somewhat laborious, limiting the sample throughput. Nonetheless, such limitations should not preclude further development in this field given that a number of steps to address these limitations have already been made with the availability of commercial standard reference materials, the constant improvements in instrumentation and the evolution of artificial intelligence. Furthermore, the extensive body of literature outlining the important biological role that these molecules play in regulating host immunity provides further impetus for identifying innovative approaches to overcoming these limitations. Deployment of SPM-based diagnostics is also likely to be complementary to those currently used in clinical practice, such as, for example, high sensitivity C-reactive protein (CRP) or erythrocyte sedimentation rates (ESR). This is because the latter biomarkers are a readout of ongoing inflammation and are therefore endpoint molecular measurements to determine whether a specific treatment has had its desired effects. By contrast, since, as discussed in the previous sections, SPM carry potent immune-regulatory activities, changes in their concentrations in biological tissues are likely to be linked with changes in immune cell behavior. Whereby, for example, increases in AT-SPM concentrations in patients treated with aspirin or an upregulation in RvT levels in patients treated with atorvastatin will lead to changes in phagocyte behavior and a downregulation of inflammatory mediator production [7,11]. Similarly, an upregulation of MCTRs in patients with bacterial infections may be linked with a potential beneficial effect of the treatment being given to these patients given that these autacoids upregulate the ability of phagocytes to clear bacteria. Furthermore, MCTRs promote tissue repair and regeneration, thereby increasing the resilience of the host to the infectious insult [10]. Consequently, diagnostics based on these mediators may be useful as either predictors or early measures of whether a given treatment is likely to limit inflammation and decrease endpoint measures such as CRP and ESR. Therefore, while acknowledging that there is still some way to go before the potential for SPM-based biomarkers can be harnessed, we look to the future with confidence that SPM may indeed be useful as novel biomarkers that will facilitate the development of personalized medicines.

## Figures and Tables

**Figure 1 biomolecules-12-00353-f001:**
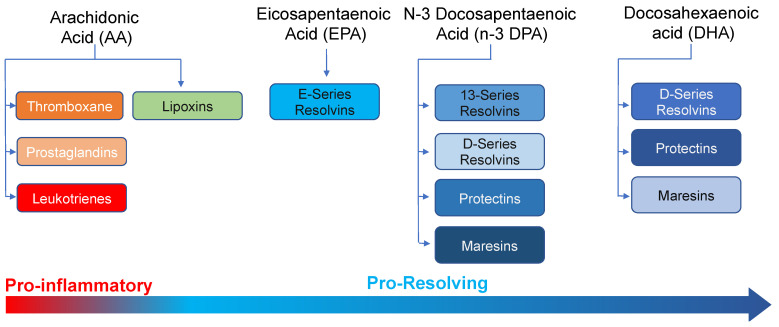
Role of lipid mediators in the initiation and resolution of inflammation.

**Figure 2 biomolecules-12-00353-f002:**
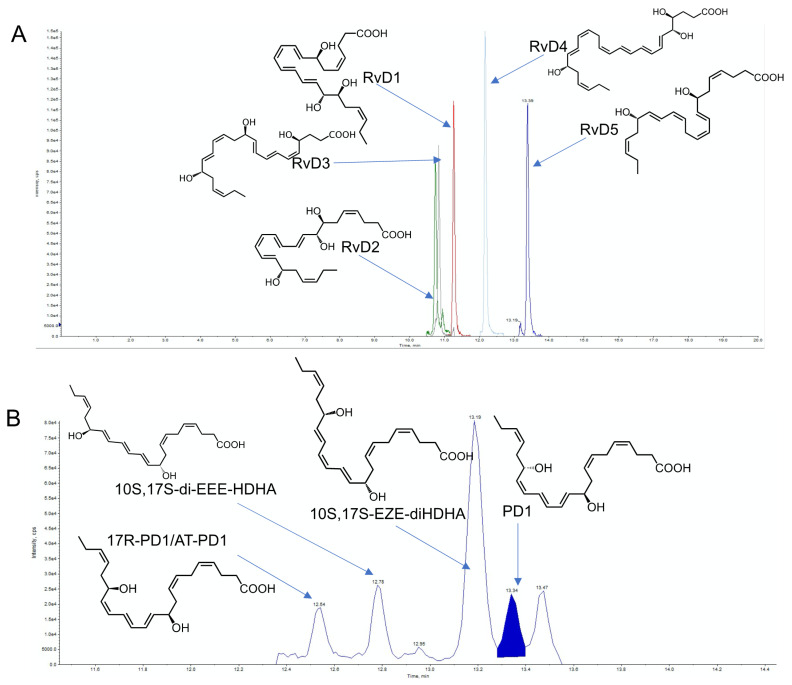
Separation of specialized pro-resolving mediators using Reverse Phase High Pressure Liquid Chromatography. Representative examples of (**A**) the chromatographic separation of DHA derived D-series resolvins demonstrating the separation of both di- and tri-hydroxylated species as well as positional isomers and (**B**) DHA-derived Protectins (PD) demonstrating the chromatographic separation of double bond and chiral isomers using RP-HPLC. Conditions employed in the separation and identification of these autacoids are as follows: samples were injected using a Shimadzu LC-20AD HPLC and a Shimadzu SIL-20AC autoinjector, paired with a QTrap 6500+ (Sciex). An Agilent Poroshell 120 EC-C18 column (100 mm × 4.6 mm × 2.7 µm) was kept at 50 °C and mediators eluted using a mobile phase consisting of methanol/water/acetic acid of 20:80:0.01 (*v*/*v*/*v*) that was ramped to 50:50:0.01 (*v*/*v*/*v*) over 0.5 min, then to 80:20:0.01 (*v*/*v*/*v*) from 2 min to 11 min, maintained until 14.5 min and then rapidly ramped to 98:2:0.01 (*v*/*v*/*v*) for the next 0.1 min. This was subsequently maintained at 98:2:0.01 (*v*/*v*/*v*) for 5.4 min, and the flow rate was maintained at 0.5 mL/min. For identification of the mediators, the QTRap 6500+ was operated in negative ion mode using a multiple reaction monitoring method with the following transitions: RvD1: *m*/*z* 375 > 141, RvD2: *m*/*z* 375 > 215, RvD3; *m*/*z* 375 > 147; RvD4: *m*/*z* 375 > 101; RvD5: *m*/*z* 359 > 199; PD1: *m*/*z* 359 > 153; 10S, 17S-EZE-diHDHA: *m*/*z* 359 > 153; 10S, 17S-EEE-diHDHA: *m*/*z* 359 > 153; 17R-PD1/AT-PD1: *m*/*z* 359 > 153.

**Figure 3 biomolecules-12-00353-f003:**
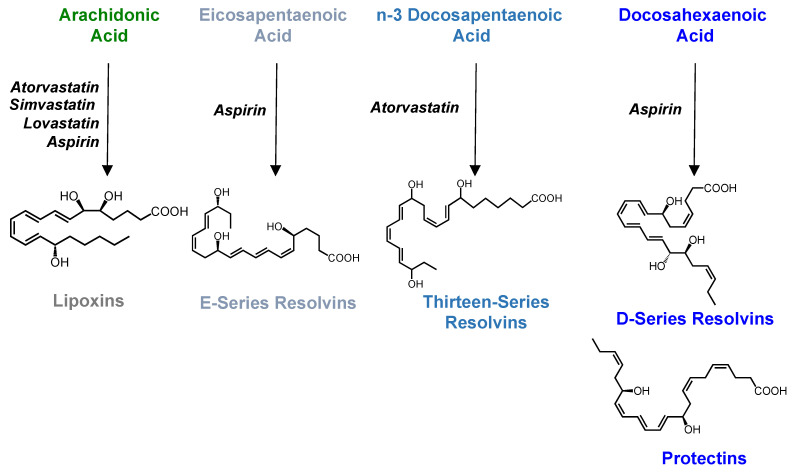
Specialized pro-resolving mediators mediate the protective activities of statins and aspirin. Mounting evidence suggests that SPM are central in mediating the protective activates of widely used drugs such as statins and aspirin. This figure provides a summary of the lipid mediator families that have been shown to mediate the anti-inflammatory activities of specific drugs.

**Figure 4 biomolecules-12-00353-f004:**
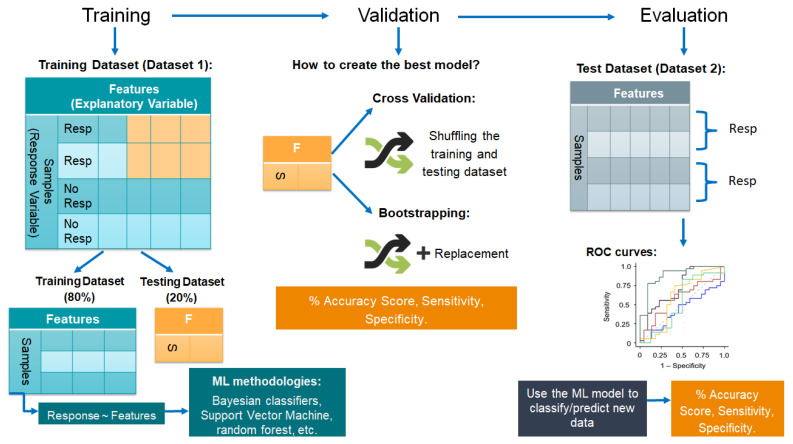
Schematic diagram of machine learning strategy. This schematic provides an overview of the distinct steps that are required for leveraging machine learning strategies for both the identification of candidate molecules and for their validation as biomarkers. The first step involves the assembly of a training dataset that contains concentrations of SPM in a cohort of patients with distinct outcomes for which a biomarker is sought. This dataset should contain a sufficient number of samples to allow for both the establishment of the dataset to be used to create the predictive models (using different machine learning strategies such us random forest) and the testing dataset. The latter should be employed to calculate how good the model is at predicting between conditions. During the validation step, the training and testing dataset are reshuffled several times using either cross-validation or bootstrapping (reshuffle with replacement) with the purpose of identifying the best predictive model and avoiding overfitting. When validation is completed, and the sensitivity and specificity of the model are calculated, biomarkers are identified, and the model can be evaluated using a new dataset. The evaluation of the model in an independent dataset is the most important step of the process since it confirms that the created model is sufficiently robust in predicting outcomes. Receiver operating characteristic (ROC) curves, among other accuracy tests, are used to evaluate the performance of machine learning models.

**Table 1 biomolecules-12-00353-t001:** Specialized pro-resolving mediator families and stereochemistries. This table provides a summary of the complete stereochemistries, where established, for the SPM distinct as subdivided by families and the essential fatty acid substrate from which they are derived. It also provides the public database reference numbers for each of these molecules where these are available.

Substrate	SPM Families	Abbreviation	Chemical Name	Lipid Maps LM ID	PubChem CID
DHA	D-series Resolvins	RvD1	7S,8R,17S-trihydroxy-4Z,9E,11E,13Z,15E,19Z-docosahexaenoic acid	LMFA04030011	44251266
		RvD2	7S,16R,17S-trihydroxy-4Z,8E,10Z,12E,14E,19Z-docosahexaenoic acid	LMFA04030001	11383310
		RvD3	4S,11R,17S-trihydroxy-5Z,7E,9E,13Z,15E,19Z-docosahexaenoic acid	LMFA04030012	71665428
		RvD4	4S,5R,17S-trihydroxydocosa-6E,8E,10Z,13Z,15E,19Z hexaenoic acid	LMFA04030002	16061138
		RvD5	7S,17S-dihydroxy-4Z,8E,10Z,13Z,15Z,19E-docosahexaenoic acid	LMFA04030003	16061139
		RvD6	4S,17S-dihydroxy-5E,7E,10Z,13Z,15E,19Z-docosahexaenoic acid	LMFA04030004	25073193
		17R-RvD1	7S,8R,17R-trihydroxy-4Z,9E,11E,13Z,15E,19Z-docosahexaenoic acid		
		17R-RvD3	4S,11R,17R-trihydroxy-5Z,7E,9E,13Z,15E,19Z-docosahexaenoic acid		
	Protectins	NPD1/PD1	10R,17S-dihydroxy-4Z,7Z,11E,13E,15Z,19Z- docosahexaenoic acid	LMFA04040001	16042541
		PDX	10S,17S-dihydroxy-4Z,7Z,11E,13Z,15E,19Z-docosahexaenoic acid	LMFA04040003	11667655
		17R-PD1	10R,17R-dihydroxy-4Z,7Z,11E,13E,15Z,19Z-docosahexaenoic acid		132282528
		22-OH-PD1	10R,17S,22-trihydroxy-4Z,7Z,11E,13E,15Z,19Z-docosahexaenoic acid		132472333
	cys-Protectins	PCTR1	16R-glutathionyl-17S-hydroxy-4Z,7Z,10Z,12E,14E,19Z-docosahexaenoic acid	LMFA04040004	132472316
		PCTR2	16R-cysteinylglycinyl-17S-hydroxy-4Z,7Z,10Z,12E,14E,19Z-docosahexaenoic acid	LMFA04040005	132472317
		PCTR3	16R-cysteinyl-17S-hydroxy-4Z,7Z,10Z,12E,14E,19Z-docosahexaenoic acid	LMFA04040006	132472318
	Maresins	MaR1	7R,14S-dihydroxy-4Z,8E,10E,12Z,16Z,19Z- docosahexaenoic acid	LMFA04050001	60201795
		MaR2	13R,14S-dihydroxy-4Z,7Z,9E,11E,16Z,19Z- docosahexaenoic acid	LMFA04050004	101894912
	cys-Maresins	MCTR1	13R-glutathionyl-14S-hydroxy-4Z,7Z,9E,11E,16Z,19Z-docosahexaenoic acid	LMFA04050005	122368871
		MCTR2	13R-cysteinylglycinyl-14S-hydroxy-4Z,7Z,9E,11E,16Z,19Z-docosahexaenoic acid	LMFA04050006	122368872
		MCTR3	13R-cysteinyl-14S-hydroxy-4Z,7Z,9E,11E,16Z,19Z-docosahexaenoic acid	LMFA04050007	122368873
n-3 DPA	13-series Resolvins	RvT1	7S,13R,20S-trihydroxy-8E,10Z,14E,16Z,18E-docosapentaenoic acid	LMFA04000091	124202379
		RvT2	7,12,13-trihydroxy-8,10,14,16,19-docosapentaenoic acid	LMFA04000092	124202381
		RvT3	7,8,13-trihydroxy-9,11,14,16,19-docosapentaenoic acid	LMFA04000093	124202383
		RvT4	7S,13R-dihydroxy-8E,10Z,14E,16Z,19Z-docosapentaenoic acid	LMFA04000094	124202385
	D-series	RvD1_n-3 DPA_	7S,8R,17S-trihydroxy-9E,11E,13Z,15E,19Z-docosapentaenoic acid		132472356
		RvD2_n-3 DPA_	7,16,17-trihydroxy-8,10,12,14,19-docosapentaenoic acid		132472324
		RvD5_n-3 DPA_	7S,17S-dihydroxy-8E,10Z,13Z,15Z,19E-docosapentaenoic acid		132472358
	Protectins	PD1_n-3 DPA_	10R,17S-dihydroxy-7Z,11E,13E,15Z,19Z-docosapentaenoic acid	LMFA04000096	132472351
		PD2_n-3 DPA_	16,17-dihydroxy-7Z,10,13,14,19-docosapentaenoic acid	LMFA04000097	132472319
	Maresins	MaR1_n-3 DPA_	7R,14S-dihydroxy-8E,10E,12Z,16Z,19Z-docosapentaenoic acid		
		MaR2_n-3 DPA_	13,14-dihydroxy-7,9,11,16,19-docosapentaenoic acid		
EPA	E-series Resolvin	RvE1	5S,12R,18R-trihydroxy-6Z,8E,10E,14Z,16E-eicosapentaenoic acid	LMFA03140003	10473088
		RvE2	5S,18R-dihydroxy-6E,8Z,11Z,14Z,16E-eicosapentaenoic acid	LMFA03140011	16061125
		RvE3	17R,18R-dihydroxy-5Z,8Z,11Z,13E,15E-eicosapentaenoic acid	LMFA03140006	60150429
		RvE4	5S,15S-dihydroxy-6E,8Z,11Z,13E,17Z-eicosapentaenoic acid		
AA	Lipoxins	LXA_4_	5S,6R,15S-trihydroxy-7E,9E,11Z,13E-eicosatetraenoic acid	LMFA03040001	5280914
		LXB_4_	5S,14R,15S-trihydroxy-6E,8Z,10E,12E-eicosatetraenoic acid	LMFA03040002	5280915
		15-epi-LXA_4_	5S,6R,15R-trihydroxy-7E,9E,11Z,13E-eicosatetraenoic acid	LMFA03040010	9841438
		15-epi-LXB_4_	5S,14R,15R-trihydroxy-6E,8Z,10E,12E-eicosatetraenoic acid	LMFA03040007	70678885

**Table 2 biomolecules-12-00353-t002:** Examples of specialized pro-resolving mediators identification in tissues from healthy volunteers and patients with inflammatory disorders.

	Disease	SPM Identified	Tissue	Method	LC Solvent System	MS/MS Ionization Mode	References
Neuronal Inflammation	Ischemic brain injury	LXA_4_	Plasma	ELISA	N/A	N/A	[74]
	Alzheimer’s disease	LXA_4_, RvD1, MaR1, 5,15-diHETE	CSF, Hippocampus	ELISA, LC-MS/MS	methanol/water/acetic acid	Negative	[75]
		RvD5, PD1, MaR1,	Entorhinal cortex tissue	LC-MS/MS	methanol/water/acetic acid	Negative	[76]
Cardiovascular diseases	Myocardial infarction	PD1, 10S,17S-diHDHA, AT-PD1, LXA_4_, AT-LXA_4_, RvD5_n-3 DPA_, PD2_n-3 DPA_, 10S,17S-diHDPA	Plasma	LC-MS/MS	methanol/water/acetic acid	Negative	[77]
		PD1, RvE1	Plasma	LC-MS/MS	Formic acid/water/acetonitrile	Negative	[78]
	Peripheral Artery Disease	LXA_4_	Plasma	LC-MS/MS	methanol/water/acetic acid	Negative	[79]
		10,17-diHDHA	Plasma	LC-MS/MS	methanol/water/acetic acid	Negative	[80]
	Atherosclerosis	RvD1, 7-epi,Δ12-trans-MaR1,	Human carotid atherosclerotic plaques	LC-MS/MS	methanol/water/acetic acid	Negative	[81]
Infections	Infection, low-dose endotoxin	RvD1, RvE1	Plasma and serum	LC-MS/MS	methanol/water/acetic acid and acetonitrile	Negative	[59]
	Periodontal disease	PD1, MaR1, LXA_4_	Saliva	ELISA	N/A	N/A	[82]
	Rhinosinusitis	RvD1, RvD2, LXA_4_	Ethmoid sinus tissue	LC-MS/MS	Formic acid/water/acetonitrile	Negative	[83]
	Tuberculosis	RvD2, 17R-PD1, PCTR3, PD1_n-3 DPA_, RvE3, 15-oxo-LXA_4_	Serum	LC-MS/MS	methanol/water/acetic acid	Positive	[84]
	Sepsis	RvD5, 17R-RvD1, PD1, 17R-PD1, RvE1, RvE2, RvE3, 5S,15S-diHETE,	Plasma	LC-MS/MS	methanol/water/acetic acid	Negative	[60]
Metabolic disease	Obesity	RvD1, RvD2, 10S,17S-diHDHA, LXA_4_, LXA_5_	Plasma	ELISA	N/A	N/A	[85]
		RvE1	Ex vivo neutrophil stimulation	LC-MS/MS	methanol/water/acetic acid	Negative	[86]
		RvD2, RvD4, LXA_4_, RvE3	Plasma	LC-MS/MS	methanol/water/acetic acid	Negative	[87]
		RvD3, RvD4, PD1, MaR1	Serum	LC-MS/MS	methanol/water/acetic acid	Negative	[88]
	Diabetes	RvD1, RvE1	Plasma	LC-MS/MS	Formic acid/water/acetonitrile	Positive	[89]
		RvD2, 17R-PD1, PCTR3, PD1_n-3 DPA_, RvE3, 15-oxo-LXA_4_	Serum	LC-MS/MS	methanol/water/acetic acid	Positive	[84]
Autoimmune disease	Rheumatoid Arthritis	RvD1, 17R-RvD1, RvD2, PD1, 10S,17S-diHDHA, MaR1, RvE1, RvE2, RvE3, 18R-RvE3	Plasma and synovial fluid	LC-MS/MS	ammonium acetate/methanol	Negative	[41]
		RvD1, RvD2, RvD3, RvD4, 10S,17S-diHDHA, 17R-PD1, MaR1, 4S,14S-diHDHA, 10S,17S-diHDPA, MaR1_n-3 DPA_, 15R-LXA_4_,	Plasma	LC-MS/MS	methanol/water/acetic acid	Negative	[40]
		PD1, LXA_4_, LXB_4_	Synovial fluid	LC-MS/MS	acetonitrile/methanol/water	Negative	[90]
	Osteoarthritis	PD1, LXA_4_, LXB_4_	Synovial fluid	LC-MS/MS	acetonitrile/methanol/water	Negative	[90]
	Hashimoto’s Thyroiditis	RvD1	Serum	ELISA			[91]
Omega-3 Supplementation	Healthy volunteers	RvD1, 17R-RvD1, RvD2, RvE1, RvE2, RvE3, 18R-RvE3	Plasma	LC-MS/MS	methanol/water/acetic acid	Negative	[92]
	Healthy volunteers and patients with periphery artery disease	LXA_4_	Plasma	LC-MS/MS	methanol/water/acetic acid	Negative	[79]
	Periphery artery disease	PDX (10S,17S-diHDHA)	Plasma	LC-MS/MS	methanol/water/acetic acid	Negative	[80]
	Major depressive disorder and chronic inflammation	RvE2, RvE3, LXB_4_	Plasma	LC-MS/MS	acetonitrile/methanol/water	Negative	[93]
	Chronic inflammation	RvD5_n-3 DPA_, MaR1_n-3 DPA_	Plasma	LC-MS/MS	acetonitrile/methanol/water	Negative	[94]

Lipid mediators denoted in red were observed to be decreased whereas those denoted in blue were observed to be increased in comparison to the respective in-study control group. (n/a = not applicable).

## Data Availability

Not applicable.

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
