# Peer review of "Utility of the Specialized Pro-Resolving Mediators as Diagnostic and Prognostic Biomarkers in Disease"

_biomolecules, 2022, doi:10.3390/biom12030353_

Round 1

Reviewer 1 Report

Dear authors,

it's possible to find also   recent reviews on the same topic ; however because there is  also a  description ( not totally correct and not well organized) of the analytical methodologies for SRM recognition I suggest  to  submit a  very well organized review on the  new and recent  analytical methodologies/ approches  to fully  characterize the SRM choosing a properly  MDPI journal based on analytics.

MINI REVIEW article

Front. Immunol., 25 November 2020 | https://doi.org/10.3389/fimmu.2020.599853

Lipid Mediators in Critically Ill Patients: A Step Towards Precision Medicine

Review Article| September 04 2020

Specialized pro-resolving mediator network: an update on production and actions

Nan Chiang ; Charles N. Serhan

Author Response

Whilst the reviewer is correct that there are other reviews on the same topics these discuss distinct aspect of SPM biology and are complementary to our article.

As to the analytical aspect, we have reviewed the document and are unaware what the reviewer is referring to as being incorrect.

Reviewer 2 Report

This is a well written review on the utility of using pro-resolving mediators as prognostic biomarkers of disease. There are minor grammatical errors and I have a couple of suggestions for inclusion within the text. Please see below for the recommended grammatical/spelling changes:

Line 48: “produces” -> produced”

Line 48: “termed as” -> “characterized as”

Line 49: “specialized pro-resolving mediators” -> “specialized pro-resolving mediators (SPM)”

Line 51: “(ALOX)” -> “(encoded by the ALOX genes)”

Line 124: “These

Lines 146 –149: please provide a reference to illustrate this statement.

Line 168 “distinguish” -> “distinguished”

Line 170: “interact the stationary phase” -> add “in” ?

Line 222: Change “form” to “from”

Line 385: “lending” -> “lend”

Line 385: add “of” between utility and peripheral

Line 463: Change to “that are better suited to the individual patient.”

Table 2 is an excellent summary of lipid mediators associated with disease. While it is not expected for the table to be exhaustive, there are relevant eicosanoids that should be included. This includes the role of 12-HETE with Type 1, Type 2 diabetes and CAD (PMID: 27353008, PMID: 12482481, PMID: 23095287; review: PMID: 34064822).

Figure 4 appears pixelated and it is very hard read the fine print. Are sex differences incorporated in this model?

Author Response

This is a well written review on the utility of using pro-resolving mediators as prognostic biomarkers of disease. There are minor grammatical errors and I have a couple of suggestions for inclusion within the text.

We would like to thank the reviewer for their time and for their helpful comments which have improved our manuscript.

Please see below for the recommended grammatical/spelling changes:

Line 48: “produces” -> produced” 11

Line 48: “termed as” -> “characterized as”

Line 49: “specialized pro-resolving mediators” -> “specialized pro-resolving mediators (SPM)”

Line 51: “(ALOX)” -> “(encoded by the ALOX genes)”

Line 124: “These

Lines 146 –149: please provide a reference to illustrate this statement.

Line 168 “distinguish” -> “distinguished”

Line 170: “interact the stationary phase” -> add “in” ?

Line 222: Change “form” to “from”

Line 385: “lending” -> “lend”

Line 385: add “of” between utility and peripheral

Line 463: Change to “that are better suited to the individual patient.”

Thank you for pointing out these typos. These are rectified in the revised manuscript.

Table 2 is an excellent summary of lipid mediators associated with disease. While it is not expected for the table to be exhaustive, there are relevant eicosanoids that should be included. This includes the role of 12-HETE with Type 1, Type 2 diabetes and CAD (PMID: 27353008, PMID: 12482481, PMID: 23095287; review: PMID: 34064822).

We thank the reviewer for the suggestion. The scope of the present review is to discuss the role of SPM as biomarkers. While we acknowledge that 12-HETE, like many other eicosanoids may be useful as potential biomarkers we believe that discussing this molecule together with lipid mediators where there is evidence for their potential utility as biomarkers is outside the scope of the present article.

Figure 4 appears pixelated and it is very hard read the fine print. Are sex differences incorporated in this model?

We have revised the figure increasing the quality of the imaging.

In these initial studies we did not include sex as one of the variables being evaluated. We have included a discussion of this on page 13 of the revised manuscript.

Reviewer 3 Report

In this paper, the authors present a comprehensive overview of the present state-of-knowledge on specialized pro-resolving mediators and their potential use in personalized medicine.

The review is nicely structured and guides the reader through a basic overview of SPM biology and their families with respective quantitation methodologies. The review also provides a brief introduction into the potential of SPMs as biomarkers in inflammatory conditions, their diagnostic potential in inflammation and for determination of therapeutic efficacy of anti-inflammatory drugs. Finally, the authors conclude the review by presenting novel machine learning strategies for evaluation of SPMs as biomarkers in personalized medicine.

The review is well-constructed, but could benefit from minor modifications/improvements.

The authors briefly mention the involvement of SPMs in resolution of allergic inflammation following treatment of dexamethasone. However, Th2 inflammation, which differs greatly from typical Th1 inflammatory conditions (such as RA), is not described before in the disease overview as well as not listed in table 2. The authors should briefly describe the identification and involvement of SPMs in those diseases as well, especially since eosinophils are characterized by one of the highest expression of ALOX15.

Same can be said for COVID-19, which is briefly listed under dexamethasone response, but the reader is not informed on the general involvement and actions of SPMs in this disease.

Figure 4 is illegible. It should be uploaded in better quality.

Author Response

In this paper, the authors present a comprehensive overview of the present state-of-knowledge on specialized pro-resolving mediators and their potential use in personalized medicine.

The review is nicely structured and guides the reader through a basic overview of SPM biology and their families with respective quantitation methodologies. The review also provides a brief introduction into the potential of SPMs as biomarkers in inflammatory conditions, their diagnostic potential in inflammation and for determination of therapeutic efficacy of anti-inflammatory drugs. Finally, the authors conclude the review by presenting novel machine learning strategies for evaluation of SPMs as biomarkers in personalized medicine.

The review is well-constructed, but could benefit from minor modifications/improvements.

We would like to thank the reviewer for their time and for their helpful comments which have improved our manuscript.

The authors briefly mention the involvement of SPMs in resolution of allergic inflammation following treatment of dexamethasone. However, Th2 inflammation, which differs greatly from typical Th1 inflammatory conditions (such as RA), is not described before in the disease overview as well as not listed in table 2. The authors should briefly describe the identification and involvement of SPMs in those diseases as well, especially since eosinophils are characterized by one of the highest expression of ALOX15.

We thank the reviewer for raising this point, we have added a section on SPM in allergic inflammation and included a section on the evidence in humans in Table 2

Same can be said for COVID-19, which is briefly listed under dexamethasone response, but the reader is not informed on the general involvement and actions of SPMs in this disease.

We thank the reviewer for raising this point, we have added a section on SPM in COVID-19 infection

Figure 4 is illegible. It should be uploaded in better quality.

We have uploaded a better quality image.

Round 2

Reviewer 1 Report

Dear Authors:

In order to make fruitful for the reader this new version of the  review ,I suggest to  prepare a table or improve the existing table adding for each indicated biomarker the analytical method used for the characterization/quantification  adding also the specific conditions; this  aspect is very import for the reader  in order that he can directly understand the applicability of the described methodology in his lab without to make vision of the connected  references, this means don't waste time and give pragmatism to a review.

Author Response

We thank the reviewer for their suggestion. We have added the requested information to Table 2.